# Assessment of the Economic and Health-Care Impact of COVID-19 (SARS-CoV-2) on Public and Private Dental Surgeries in Spain: A Pilot Study

**DOI:** 10.3390/ijerph17145139

**Published:** 2020-07-17

**Authors:** Cintia Chamorro-Petronacci, Carmen Martin Carreras-Presas, Adriana Sanz-Marchena, María A Rodríguez-Fernández, José María Suárez-Quintanilla, Berta Rivas-Mundiña, Juan Suárez-Quintanilla, Mario Pérez-Sayáns

**Affiliations:** 1Unit of Oral Medicine, Faculty of Medicine and Dentistry, University of Santiago de Compostela, 15705 Santiago de Compostela, Spain; cintia.chamorro.petronacci@gmail.com; 2Health Research Institute of Santiago de Compostela (IDIS), 15706 Santiago de Compostela, Spain; 3Professional Association of Dentists and Stomatologists of Madrid, 28046 Madrid, Spain; carmen.martin2@universidadeuropea.es; 4Presidency of the Professional Association of Dentists and Stomatologists of Pontevedra-Ourense, 36003 Pontevedra, Spain; adriana.sanz.marchena@gmail.com; 5Department of preventive medicine and public health, University of Santiago de Compostela, 15705 Santiago de Compostela, Spain; almudena.rodriguez@usc.es; 6Presidency of the Professional Association of Dentists and Stomatologists of A Coruña-Lugo, 15011 A Coruña, Spain; josemaria.suarez@usc.es; 7Pathology and Therapeutic Unity, Faculty of Medicine and Dentistry, University of Santiago de Compostela, 15705 Santiago de Compostela, Spain; berta.rivas@usc.es; 8Galician Public Health Service (SERGAS), Dentistry and Primary Health Care of the Health District of Santiago de Compostela, 15705 Santiago de Compostela, Spain; juanantonisuarez.suarez@usc.es

**Keywords:** COVID-19, SARS-CoV-2, economic impact, dental office management

## Abstract

Objectives: The COVID-19 (SARS-CoV-2) pandemic is an ongoing public health challenge, also for the dentistry community. The main objective of this paper was to determine the economic and health-care impact of COVID-19 on dentists in the Autonomous Region of Galicia (Spain). Methods: This was a descriptive observational study in which the data was collected by means of a self-administered survey (from 1 April 2020 to 30 April 2020). Results: A total of 400 dentists from Galicia responded to the survey. Only 12.3% of the participants could obtain personal protective equipment (PPE) including FFP2 masks. Of the male respondents, 33.1% suffered losses >€15,000 compared to 19.4% of female respondents (OR = 3.121, *p* < 0.001). Economic losses seem to have contributed to the applications for economic help as 29.5% of the respondents who applied for this measure recorded losses in excess of €15,000 (*p* = 0.03). Patients complained more about the fact that only emergency care was available during the State of Alarm, in dental surgeries that do not work with insurance companies or franchises. Only 4 professionals tested positive, 50% of whom worked exclusively in private practice and the other 50% who practised in both private and public surgeries. Dentists who practise in the public sector saw more urgent patients per week than those practising in private surgeries (*p* = 0.013). Conclusions: The COVID-19 pandemic has had economic repercussions in dentistry as only urgent treatment was available during the State of Alarm. These repercussions seem to be higher in male participants, as the majority of the participants have revealed higher economic losses than females. The level of assistance has also been affected, reducing the number of treated patients, although this quantity has been different in private and public surgeries. By presenting these findings we look to highlight the role that dentists play in society in treating dental emergencies in our surgeries, and this must be recognised and addressed by the relevant authorities, who must provide PPEs as a priority to this group as well as providing special economic aid in accordance with the losses incurred by the sector.

## 1. Introduction

On the 11th of March, 2020, COVID-19 (SARS-CoV-2) was declared as a pandemic by the World Health Organization (WHO), and it has been an ongoing public health challenge in more than one hundred countries [1]. The first case in Spain was detected at the end of January 2020, however, the State of Alarm (Spanish Royal Decree 14/3/2020) was not declared until the 14th of March. In order to face this health emergency, the population’s freedom of movement was limited and all activities in the commercial, recreational and hospitality sectors were suspended. Dental surgeries were not forced to close, nevertheless, their activity was limited exclusively to emergency treatments by the Spanish Order SND/310/2020 (Ministry of Health), issued on the 31st March.

According to institutions and professional associations, this new situation has been the causal factor for the economic and health care-related difficulties which have affected the dental sector. The difficulties which this sector has faced are the result of the limited access to personal protective equipment (PPE), given that this equipment was exclusively allocated to public institutions, and the complexity in managing the financial aid as approved by the new legislation, which was passed during the State of Alarm.

Dentists are one of the professional groups that work in very frequent contact with patients, and it is not possible for them to maintain a safe distance while performing dental treatments, as a result, this collective has a higher probability of being infected by different micro-organisms, such as COVID-19 [2]. The majority of the guidelines and protocols of action established for dental practices for the COVID-19 pandemic emphasise the high risk of transmission of the virus during dental treatment, mainly due to the frequent production of aerosol and direct contact with the patient’s saliva [3,4,5,6]. Dental professionals, like all other health workers, also have an ethical and deontological duty to attend to situations which are categorised as urgent, as by doing so, they can prevent more strain from being put on the hospital emergency services and stop healthy patients from being infected while waiting for their dental treatment in the hospital [7].

Considering the delay in declaring the State of Alarm (a month and a half after the first positive case was confirmed), we consider the situation for dental surgeries during the COVID-19 crisis to be of great interest. Furthermore, the equipment protocols for routine dental practice do not include the use of FFP2 masks (masks with 92% minimum filtration efficiency) which seem to be necessary to prevent the virus from spreading [8].

Social distancing and hygiene measures seem to be the most effective methods for controlling the risk of COVID-19 cross-infection. Given the specific characteristics of the COVID-19 virus (a fast-mutating RNA), the need for further lockdowns in the future, which would lead to a decline in economic activity, cannot be ruled out [9].

Although the majority of the publications, institutions, guidelines and public organizations acknowledge that the COVID-19 pandemic has an economic and health-care impact on the dental sector [10], nevertheless, the majority of studies on this pandemic address factors such as the fear of contagion, knowledge about the particular aspects of the virus or prevention protocols [2,11,12].

In order to be able to propose safe guidelines and procedures for both patients and professionals, we must firstly be aware of how dental surgeries have been affected by COVID-19, and for this reason, the main objective of this paper was to determine the economic and health-care impact that the COVID-19 pandemic has had on dentists in the Autonomous Region of Galicia.

## 2. Materials and Methods

This is a descriptive observational study for which the data was collected by means of a self-administered survey. The study was aimed at practicing dentists who were working when the State of Alarm was declared (Royal Degree 14/0/2020, norm with rank of law) in Spain. This study was conducted following the STROBE (Strengthening the Reporting of Observational studies in Epidemiology) guidelines for observational studies [13] and was evaluated by the Galician Ethics Committee (Ref. IDIS/2020/COVID19-ODONTO).

The questions for this study were drawn up after revising the relevant literature on COVID-19 [14,15,16]. The questionnaire was conducted in Spanish and was initially tested by 20 professionals in order to evaluate the respondents’ understanding of the questions and to determine how long it took them to complete the questionnaire.

The questionnaires were distributed (from 1 April 2020 to 30 April 2020) by the Professional Associations of the regions of A Coruña, Lugo, Ourense and Pontevedra to dentists who worked in Galicia Community (Region of Northwest of Spain) through newsletters and posts on their respective websites.

The potential of our study was calculated using Epidat 4.2 software (SERGAS, Galicia, Spain). Out of a total sample of 2020 dentists, the response rate was 19.95%, with a 50% heterogeneity and a 5% margin for error, therefore a 97% confidence level was obtained.

The data was statistically analysed using the SPSS v.24.0 (IBM, Statistics, Armonk, NY, USA) program. The categorised variables were described using frequencies and percentages. Contingency tables were established in order to study the relationship between the categorical variables using the Chi-square test, and the degree of association was determined using Cramer’s V. In order to assess the risk (OR) of the principal variables, multinomial logistic regression was used and this was done by recoding and grouping the variables. The significance value was established as *p* ≤ 0.05.

## 3. Results

A total of 400 dentists from Galicia responded to the survey. The distribution of the responses is shown in Table 1. The majority of the participants were women (64.5%), and half of our sample group was 45 years old or younger.

Economic losses seem to have contributed to the applications for TERF (temporary employment regulation file), as 29.5% of the respondents who applied for this measure recorded losses in excess of €15,000 (Chi^2^ = 10.744, Cramer’s V= 0.116, *p* = 0.03). For losses exceeding €15,000, the OR for applying for a TERF was 3.248 (CI95% 0.976–10.805, *p* = 0.05) compared with an OR for not applying for a TERF of 1.895 (CI95% 0.189–19.039, *p* = 0.587). With regards to the approval of the TERF, the OR for surgeries which did not apply for this measure, or whose request was rejected, was higher in the cases of surgeries with losses of less than €15,000 (OR = 4.545, CI95% 1.425–14.504, *p* = 0.011), compared to those with higher losses, in which the OR for being rejected was lower (OR = 3.433, CI95% 2.024–5.822, *p* < 0.001).

33.1% of the male respondents suffered losses >€15,000 compared to 19.4% of female respondents (OR = 3.121, CI95% 1.715–5.679, *p* < 0.001). In the lower income ranges (€1000–4999), 32.2% of the female respondents suffered these losses compared with 17.6% of the male respondents (Chi^2^ = 16.938, Cramer’s V = 0.206, *p* = 0.002) (Figure 1). No significant differences were revealed in our analysis of economic losses according to specialty.

The percentage of patients that complained about the fact that only emergency care was available during the State of Alarm (>10% of the patients as the reference value) was higher in dental surgeries that do not work with insurance companies or franchises, 75% vs 25% in franchise surgeries, (Chi^2^ = 9.848, Cramer’s V = 0.157, *p* = 0.002). The OR for high patient complaints (>10%) in non-franchise surgeries was 2.943 (CI95% 1.463–5.918, *p* = 0.002).

Only 4 professionals tested positive, 50% of whom worked exclusively in private practice and the other 50% who practiced in both private and public surgeries. The respondents who were not tested or who tested negative, mainly worked exclusively in private practice, 90.2% (Chi^2^ = 6.940, Cramer’s V = 0.132, *p* = 0.008). The OR for testing negative for COVID-19 among those working exclusively in private practice was 9.154 (CI95% 1.254–66.808, *p* = 0.029).

Dentists who practice in the public sector saw more urgent patients per week than those practicing in private surgeries (Chi^2^ = 12.749, Cramer’s V = 0.126, *p* = 0.013) (Figure 2). The OR for seeing more than 11 urgent patients per week was 6.270 for public surgeries (CI95% 1.933–20.339, *p* = 0.002).

## 4. Discussion

A total of 400 dentists responded to this survey, the majority of whom (64.5%) were women, with the highest levels of participation from the 36–45 age range (27%). The response rate was low, however, this was probably due to a lack of motivation amongst professionals given the harshness of the situation. In terms of gender, our participation results are similar to previous remote and/or physical surveys, where the percentage of female participation ranges from 66–75% [17,18]. The majority of our respondents (89.9%) carry out their activity in a private dental surgery, and this is similar to the results obtained in other consulted surveys about COVID-19 (74%) [18]. The significant link between gender and specialization can be seen in oral surgery (29.6% men vs 13.2% women) and in paedodontics (4.3% women vs 0.7% men) and orthodontics (11.2% women and 5.6% men). This is in line with the results of other studies that reveal certain preferences among men and women for specializations [19].

Despite equality policies, the loss of purchasing power amongst the female gender, seems to be perpetuated over time [20]. Some studies suggest that women reduce their working hours when they have children. Given these economic repercussions, it seems reasonable that the difference in loss should be significant between the two groups [21].

1% of respondents confirmed that they had tested positive for COVID-19. Considering the high risk involved in administering any dental treatment, it seems that Spanish dentists have followed the guidelines and recommendations of the different institutions with sound judgement [22]. It is evident that there has been a decrease in the number of patients seen during the State of Alarm, with the majority of the respondents (85.5%) seeing between 1–5 patients per week following the recommendations to treat only urgent situations [6,23].

PPEs seem to be essential in order to be able to provide care in a safe manner and to control cross-infection [24]. According to our data, more than 60% of the respondents were unable to obtain any type of protective material, and only 12.3% were able to obtain this material including FFP2 masks. This data coincides with the survey carried out by the SESPO (Spanish Society of Oral Public Health) [25]. In our case, there were no differences between those surveyed in the first half of April and in the second half of the same month (*p* = 0.506) regarding the acquisition of PPEs, therefore showing that there were no improvements in terms of the provision of this material over the course of the month. We believe that this was due to the fact that public health care units had priority to receive PPEs.

The Spanish Government has taken some measures to reduce the economic impact of COVID-19 in the most affected sectors. The economic losses for the respondents were, for the most part, (27%) between 1000–4999 euros, followed by 5000–9999 euros (25.5%). The comparative analysis reflects a higher rate of TERFs requested in the cases in which the economic loss was greater (*p* < 0.01), as well as the favorable response to said requests (*p* < 0.01). In order to reduce the economic crisis, those companies forced to close down or reduce their working hours could benefit from the TERF, justified through technical, organizational or production reasons. During this situation, even though the worker does not work or reduced her or his working day, the payroll and the contribution is assumed by the State.

Although we have found papers that recorded the loss of patients and revenue in dental surgeries due to the impact of SARS CoV-1 (16%) [26], specifically in the Taiwan area, we have not yet found any papers that address the economic impact of COVID-19.

With regards to the patients’ attitude to the implementation of the State of Alarm and special dental care situation, the majority of the respondents (86%) stated that none of their patients or less than 10% had complained about it, although 10% of the respondents stated that 10–40% of their patients had expressed their dissatisfaction. On the other hand, 28% of the respondents stated that 10–30% of their patients requested an appointment for what the dentist considered as non-emergency treatment. Likewise, the respondents who perform their activity in non-franchise dental surgeries or do not work with dental insurance companies reported significantly different patient responses (*p* = 0.008). All the respondents who claimed that “more than 70% of the patients had complained about this new situation”, work in non-franchise dental surgeries. We think that these results must be interpreted cautiously. Franchise dental surgeries have a different organization where most of the time there is a receptionist or an auxiliary that deals with patient’s complaints or triage, while in the non-franchise dental surgeries (and taking into account the possibility that the owner has sent most of their personal staff to TERF) the dentist can perform this task. This means that patient complaints may be similar but, as we have only asked dentists, it is likely that the work of registering them in some cases has been done by another staff member.

One of the limitations of this study was the exclusive use of self-administered online surveys. Another limitation was the time in which this survey was available (from 1 April 2020 to 30 April 2020), and the fact that when the survey was completed, the State of Alarm was still in force in Spain, therefore meaning that the economic losses may be greater than those assessed in this survey. Another limitation of this study is that it did not consider any economic aids other than TERFs, and that TERF has only been analyzed as an economic factor, when it also has assistance repercussions, as the company has less workers for patient care.

This is a pilot study, and in future research our group intend to increase the sample size to the whole territory of Spain, add questions such as knowledge about COVID-19 in the dentistry community, risk perception and fear on treating patients. Discussions about future challenges in dentistry include new situations as dental tele assistance, which should be limited scientifically and deontologically. Adjustment for this new reality is unavoidable and these new aspects require consideration for dental professionals and patients.

## 5. Conclusions

The COVID-19 pandemic has had economic repercussions in dentistry as only urgent treatment could be available during the State of Alarm. These repercussions seem to be higher in male participants, as the majority of them have revealed higher losses than females. The level of assistance has also been affected, reducing the number of treated patients, although this quantity has been different in private and public surgeries. By presenting these findings we highlight the role that dentists play in society in treating dental emergencies in our surgeries, and this must be recognized and addressed by the relevant authorities, who must provide PPEs as a priority to this group as well as providing special economic aid in accordance with the losses incurred by the sector.

## Figures and Tables

**Figure 1 ijerph-17-05139-f001:**
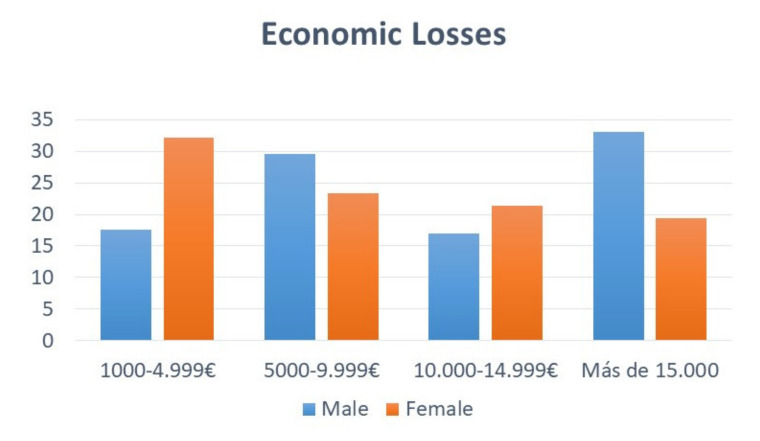
Economics of female and male losses.

**Figure 2 ijerph-17-05139-f002:**
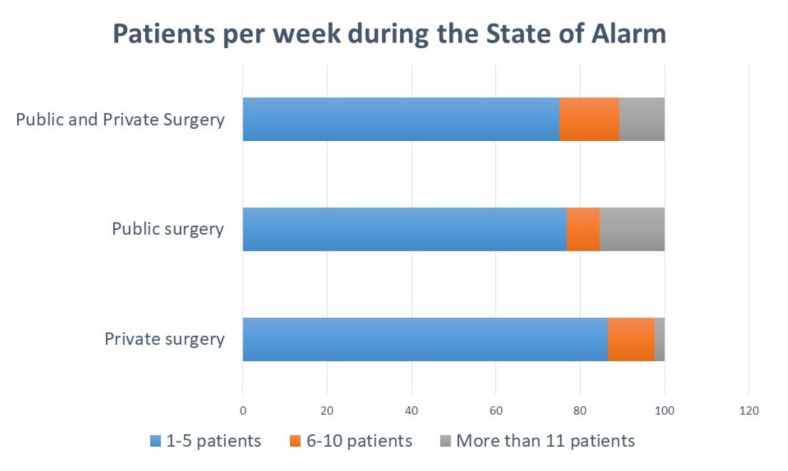
Patients per week during the State of Alarm in public and/or private surgery.

**Table 1 ijerph-17-05139-t001:** Responses distribution.

Questionnaire Items	No. (%)
Sex	
Male	142 (35.5)
Female	258 (64.5)
Age	4 (1)
Between 22–25	108 (27)
Between 26–35	138 (34.5)
Between 26–35	79 (19.8)
Between 56–65	67 (16.8)
65 and older	4 (1)
Type of surgery where you work:	
Exclusively private practice	359 (89.9)
Exclusively public practice	13 (3.3)
Public and private practice	28 (7)
Do you work in more than one surgery?	
No	218 (54.5)
Yes	182 (45.5)
How many?	
2	98 (24.5)
3-4	58 (14.5)
More than 4	24 (6)
Are any of these surgeries franchises or dental insurance companies?	
No	351 (37.8)
Yes	49 (12.3)
Average number of patients seen per week:	
1–20	25 (6.3)
21–40	98 (24.5)
41–60	140 (35)
More than 60 patients	137 (34.3)
Speciality:	
Oral Surgery and Implantology	76 (19)
Endodontics	24 (6)
General Dentistry	241 (60.3)
Paedodontics	12 (3)
Orthodontics	37 (9.3)
Prosthodontics	10 (2.5)
Before the State of Alarm was declared (14th March) did you use FFP2 masks in routine dental practice?	
No	383 (95.8)
Yes	17 (4.3)
Did you manage to procure PPE (Personal Protective Equipment) for yourself and your colleagues once the State of Alarm had been declared?	
I was unable to procure any PPE	243 (60.8)
Yes, I managed to procure caps, FFP2 masks and disposable gowns	44 (11)
Yes, I managed to procure caps and masks	26 (6.5)
Yes, I managed to procure caps or masks	38 (9.5)
Yes, I managed to procure caps, FFP2 masks and disposable gowns.	49 (12.3)
Regarding COVID-19 test:	
I was not tested	385 (96.3)
I tested negative	11 (2.8)
I tested positive	4 (1)
If you tested positive, what type of symptoms did you have?	
Mild symptoms	8 (2)
Serious symptoms (required hospitalisation)	0
Did any of your employees or non-dentist colleagues (receptionists, dental hygienists or clinic assistants) test positive for COVID-19?	
Some of those who were tested came back positive	9 (2.3)
No one tested positive	18 (4.5)
No one was tested	373 (93.3)
Do you know any dentist who tested positive for COVID19?	
No	300 (75)
Yes	100 (25)
Do you know any dentist who died from COVID19? No Yes	390 (97.5) 10 (2.5)
What percentage of patients called to cancel or informed of their intention to cancel previously scheduled appointments after the State of Alarm was declared on the 14th of March?	
Less than 10%	344 (86)
10–40%	41 (10.3)
40–70%	12 (3)
More than 70%	3 (0.8)
How many patients did you see per week during the State of Alarm?	
1–5 patients/week	342 (85.5)
6–10 patients/week	45 (11.3)
More than 11 patients/week	13 (3.3)
Did any of the patients request treatment for a situation which they considered to be a medical emergency but that you did not consider as such?	
No, or less than 10%	254 (63.5)
Yes, between 10–30%	112 (28)
Yes, between 31 and 60%	22 (5.5)
More than 60%	12 (3)
What percentage of patients expressed their dissatisfaction regarding the provision of dental care exclusively for emergency cases?	
Less than 10%	344 (86)
10–40%	41 (10.3)
40–70%	12 (3)
More than 70%	3 (0.8)
If you are the owner of the dental surgery, did you apply for a TERF (temporary employment regulation file) for any of the dental surgery’s staff?	
No	43 (10.8)
Yes	219 (54.8)
I am not the owner	138 (34.4)
If you applied for a TERF, was it approved by the government?	
No	19 (4.8)
Yes	225 (56.3)
I did not apply for it	156 (39)
If you work at a private surgery, what economic losses have been caused by the COVID-19 pandemic in terms of monthly income?	
1,000–4,999 €	108 (27)
5,000-9,999 €	102 (25.5)
10,000–14,999 €	79 (19.8)
More than 15,000 €	97 (24.3)
I do not work at a private surgery	14 (3.5)
Have any of the surgeries where you work closed down?	
No	348 (87)
Yes	52 (13)

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
