# Peer review of "Assessment of the Economic and Health-Care Impact of COVID-19 (SARS-CoV-2) on Public and Private Dental Surgeries in Spain: A Pilot Study"

_ijerph, 2020, doi:10.3390/ijerph17145139_

Round 1
Reviewer 1 Report
Comments to the Author
Dear authors, thank you for this interesting MS that highlights the COVID-19 and economic. Please note the following points that need your attention:
Introduction:
P2, Line 55: What does that mean, SND/310/2020. Mistyped?
You should explain more about ERTE.
P2, Line 75: What do you stand for FFP2. You should explain about FFP2 mask.
Material & Methods:
P2, Line 91: What does that mean, R.D.14/0/2020. Mistyped? What do you stand for R.D.
P3, Line 98: Where did you send the questionnaires. You should explain the area of destination.
P3, Line 109: Why was not the significant value “p<0.05”? Why did you include p=0.05 as the significant value?
Results:
P6, Line 135: In Line 133, You described that “The percentage of patients that complained was lower in franchise surgeries”. But, the OR for low patient complaints in non-franchise surgeries was 2.943. Was the percentage of patients that complained higher in non-franchise? If so, the OR for low patient complains in non-franchise surgeries is less than 1.
Discussion:
You should discuss more about ERTE’s results. Your limitation described that you consider only the ERTE as economic factor.
P7, Line 190: In your country, What is the ration between patients booked and patients not booked? If you consider about the ratio of appointment canceled. You should explain your cultural background about dentist’s appointment.
P7, Line 193: What is the results p=0.008. Please, explain more about significantly different.
Thank you.
Author Response
Reviewer 1:
Introduction: P2, Line 55: What does that mean, SND/310/2020. Mistyped?
The acronym SND in Spain means that it comes from the Ministry of Health. We clarify this adding “Spanish order… (Ministry of Health)”, in page 2 line 54 and 55. The complete order is available at: https://www.boe.es/buscar/act.php?id=BOE-A-2020-4211.
You should explain more about ERTE.
ERTE comes from Spanish initials for the term Temporary Employment Regulation File (TERF). We have translated the terms to English and change all the ERTE to TERF. We have added an explanation in lines 186-190: “In order to reduce the economic crisis, those companies forced to close down or reduce their working hours could benefit from the TERF, justifying causes technical, organizational or production reasons. During this situation, even though the worker does not work or reduced her or his working day, the payroll and the contribution is assumed by the State”.
P2, Line 75: What do you stand for FFP2. You should explain about FFP2 mask. We have added an explanation about FFP2 characteristics in line 75: “masks with 92% minimum filtration efficiency”.
Material & Methods: P2, Line 91: What does that mean, R.D.14/0/2020. Mistyped? R.D. are acronyms for Spanish term “Real Decreto”. It has been translated to English by Royal Degree, which is a legal norm with rank of law, and we have clarified this in line 93.
P3, Line 98: Where did you send the questionnaires. You should explain the area of destination.
Questionnaires were distributed to Dentists in the Galician Community (Region of Northwest of Spain), this information has been added in lines 101 and 102.
P3, Line 109: Why was not the significant value “p<0.05”? Why did you include p=0.05 as the significant value?
The level of significance was considered as p values less or equal to 0.05 and it has been indicated in material and methods, page 3, line 113. Authors have been based on the widespread use of “statistical significance” (generally interpreted as “p ≤ 0.05”) as a license for making a claim of a scientific finding. (Wasserstein RL, Lazar NA (2016). «The ASA's statement on p-values: context, process, and purpose». The American Statistician. doi:10.1080/00031305.2016.1154108).
Results: P6, Line 135: In Line 133, You described that “The percentage of patients that complained was lower in franchise surgeries”. But, the OR for low patient complaints in non-franchise surgeries was 2.943. Was the percentage of patients that complained higher in non-franchise? If so, the OR for low patient complains in non-franchise surgeries is less than 1.
Effectively, this was a mistake during the translation and is not “low patient complains in non-franchise surgeries”, but “high patient complains in non-franchise surgeries… > 10%”. This has been corrected in line 138.
Discussion: You should discuss more about ERTE’s results. Your limitation described that you consider only the ERTE as economic factor.
This is an interesting point of view, and we have decided to add more information about the other possible repercussions of TERF (previously ERTE). If a company take TERF means that it has less workers and less assistance capacity. It is interesting to evaluate the impact on dental care due to the reduction in staff from the perspective of the dentist and the patient. We have added this explanation in lines 217-218.
P7, Line 190: In your country, what is the ration between patients booked and patients not booked? If you consider about the ratio of appointment cancelled. You should explain your cultural background about dentist’s appointment.
This information is unknown to the authors and we have not found any solid bibliographic reference to discuss these results.
P7, Line 193: What is the results p=0.008. Please, explain more about significantly different.
We have clarified these results in lines 201-208, and we think that now is more understandable: “All the respondents who claimed that “more than 70% of the patients had complained about this new situation”, work in non-franchise dental surgeries. We think that these results must be interpreted cautiously because franchise dental surgeries have different organization where most of the times there is a receptionists or an auxiliary that deal with patient´s complaints or triage, while in the non-franchise dental surgeries (and taking into account the possibility that the owner have sent most of their personal staff to TERF) the dentist can perform this task. We want to say that patient complaints may be similar but, as we have only asked dentists, it is likely that the work of registering them in some cases has been done by another staff member”.
Reviewer 2 Report
The study is a short report of findings from a Covid-19 survey among dentists. As such, it is professionally performed and lives up to what can be requested from a statistical viewpoint, and it is of utmost interest as new evidence in extension of the pandemic.
One could ask for more sophisticated statistical analyses. However, given the sample size at hand, it is advisable to restrict to the narrow approaches taken by the authors.
The only major weakness of the study is that it lacks a perspective on future research. Specifically, the authors should devote a part of the discussion to potential shortcomings of the present study and its results, with prospective recommendations of what to take up in future research. Are there further questions that should be taken up? Would it pay off to increase the sample size? Etc...
Such a discussion would add to the futrure impact of the study. If it is added, then I will recommend accept.
Author Response
We greatly appreciate the considerations made by the reviewer, in fact, this study is a pilot study of a much larger project involving all the staff of dentistry in Spain. This new and wide project includes the 40.000 dentists that work in our country and is formed by a multidisciplinary group of professionals, that include dentists, specialists in law, and economists. This project is currently under evaluation and pending funding, by the foundation of the Conference of Rectors of Spanish Universities Universities (CRU, Ref: EVASECO/2020). Our main objective is to deepen in the economic and welfare impact in the different regions of Spain and in the control and transmission mechanisms in which dentists are involved when it comes to preventing possible outbreaks of the COVID-19 pandemic.
We have considered this comment very useful and we have added in the discussion our perspective for a future research (lines 218-223):
“This is a pilot study, and in future research our group intend to increase the sample size to the whole territory of Spain, add questions such as knowledge about COVID-19 in dentistry community, risk perception and fear on treating patients. Discussion about future challenges in dentistry include new situations as dental tele assistance, which should be limited scientifically and deontological. Adjustment for this new reality is unavoidable and these new aspects require consideration for dental professionals and patients”.
Reviewer 3 Report
The presented publication takes up a current hypothesis that COVID-19 (SARS-CoV-2) pandemic has been an ongoing public health challenge also for dentistry community.
The main objective of this paper was to determine the economic and health-care impact of COVID-19 on dentists in the Autonomous Region of Galicia (Spain). This was a descriptive observational study in which the data was collected by means of a self-administered survey (from 1st April, 2020 to 30th April, 2020). A total of 400 dentists from Galicia responded to the survey. Only 12.3% of the participants could obtain protective material (PPEs) including FFP2 masks. 33.1 % of the male respondents suffered losses >€15,000 compared to 19.4% of female respondents (OR=3.121, p<0.001). Economic losses seem to have contributed to the applications for economic helps as 29.5% of the respondents who applied for this measure recorded losses more than €15,000 (p=0.03).
The aim of the publication is clearly visible. Since the reviewer himself is also a dentist in private practice, he can only express his unconditional support for the incredibly difficult situation of his colleagues from Galicia because of the Corona crisis.
Unfortunately, this is not the task of a review. From a scientific point of view, the reviewer must unfortunately note that although the standardized requirements for a questionnaire-based observational study have been complied with, but the informative value of the publication is significantly reduced
- by the comparatively low number of responders not qualifying the results as representative for the Galicia dentistry,
- the one-sided focus on mainly young practice owners or employees in state institutions, which is clearly recognizable from the answers,
and
- the inadequate validation of the questionnaire (based exclusively on a small group of experts), whose qualifications are unfortunately not proven.
In conclusions, it is no doubt, the COVID-19 pandemic has had economic repercussion in dental dentistry during the State of Alarm due to this pandemic. We need urgently publications presenting statistically confirmed true findings to highlight the role that dentists play in society considering the COVID-9 pandemic. The results of these publications must be recognized and addressed by the relevant authorities, who must provide PPEs as a priority to the dentist as an important sector of medical aid as well as providing special economic aid in accordance with the losses incurred by the sector.
Unfortunately, this particular publication is not fulfilling the scientific demands on for being published in this journal. For this reason, the reviewer believes that the manuscript cannot be released for publication.
Author Response
We would like to thank the reviewer for his empathy with the dental collective and appreciate his comments regarding the manuscript. We are aware that this work has a great limitation in sample size due to the accessibility to all dentists by Data Protection Law. Even with these limitations, we have managed to reach a quarter of the target population. Considering that this is the first work known to date on the economic and health aspects and impact on health, we consider that the novelty of the work, can compensate this pilot study a smaller sample size.
Maintaining the emergency service in the dental field is essential to reduce the number of patients who come to the health system and hospitals that are already overloaded. Our work shows a perspective of dentists in the private and public sphere in the north of Spain, in addition to the measures implemented to minimize the risk of transmission, the challenges that dentists have faced and the impact it has had on our patients.
This work allows scientific community compare publications of other countries and with other policies and perspectives, such as the Italian or Canadian one, although economic valuation is an added potential of our work that we have not found in other similar ones (PMID: 32472974; PMID: 32481672).
Our work has obvious limitations, such as the aforementioned limited sample, or the fact that the survey was carried out during the pandemic itself and not after its resolution. However, we believe that the information provided by this survey on the economic and healthcare impact of COVID-19 during this state of exception is very useful for the scientific and health community. Anyway, we have to continue working on aspects such as virus transmission, the implementation of new protocols that allow remote assistance and discuss aspects such as privacy and patient consent. And so we have reflected these future intentions in lines 218-223 in Discussion section.
Considering all of the above, we will reconfigure the manuscript and consider it as a pilot and novel study to support future research. We have added a modification to the title, reflecting that it is a pilot study (line 4).
Round 2
Reviewer 1 Report
Thank you for your good work. Wishing you all the best.
Reviewer 3 Report
The presented publication takes up a current hypothesis that COVID-19 (SARS-CoV-2) pandemic has been an ongoing public health challenge also for dentistry community.
The main objective of this paper was to determine the economic and health-care impact of COVID-19 on dentists in the Autonomous Region of Galicia (Spain). This was a descriptive observational study in which the data was collected by means of a self-administered survey (from 1st April, 2020 to 30th April, 2020). A total of 400 dentists from Galicia responded to the survey. Only 12.3% of the participants could obtain protective material (PPEs) including FFP2 masks. 33.1 % of the male respondents suffered losses >€15,000 compared to 19.4% of female respondents (OR=3.121, p<0.001). Economic losses seem to have contributed to the applications for economic helps as 29.5% of the respondents who applied for this measure recorded losses more than €15,000 (p=0.03).
The aim of the publication is clearly visible. Since the reviewer himself is also a dentist in private practice, he can only express his unconditional support for the incredibly difficult situation of his colleagues from Galicia because of the Corona crisis.
The changes, insertions and corrections made by the authors in the revised version of the publication are convincing. The shortcomings of the publication as presented by the reviewer in his review from 25.06.2020 have been corrected and supplemented accordingly by the authors. Therefore, the reviewer can now agree to a publication.
